# Extracellular Vesicles from Thyroid Carcinoma: The New Frontier of Liquid Biopsy

**DOI:** 10.3390/ijms20051114

**Published:** 2019-03-05

**Authors:** Germana Rappa, Caterina Puglisi, Mark F. Santos, Stefano Forte, Lorenzo Memeo, Aurelio Lorico

**Affiliations:** 1College of Medicine, Touro University Nevada, 874 American Pacific Drive, Henderson, NV 89014, USA; germana.rappa@tun.touro.edu (G.R.); mark.santos@tun.touro.edu (M.F.S.); 2Mediterranean Institute of Oncology Foundation, Via Penninazzo 7, 95029 Viagrande, Italy; caterina.puglisi@grupposamed.com (C.P.); stefano.forte@grupposamed.com (S.F.); lorenzo.memeo@grupposamed.com (L.M.)

**Keywords:** thyroid carcinoma, papillary thyroid carcinoma, liquid biopsy, cancer, extracellular vesicles, exosomes, diagnostic biomarkers, miRNA

## Abstract

The diagnostic approach to thyroid cancer is one of the most challenging issues in oncology of the endocrine system because of its high incidence (3.8% of all new cancer cases in the US) and the difficulty to distinguish benign from malignant non-functional thyroid nodules and establish the cervical lymph node involvement during staging. Routine diagnosis of thyroid nodules usually relies on a fine-needle aspirate biopsy, which is invasive and often inaccurate. Therefore, there is an urgent need to identify novel, accurate, and non-invasive diagnostic procedures. Liquid biopsy, as a non-invasive approach for the detection of diagnostic biomarkers for early tumor diagnosis, prognosis, and disease monitoring, may be of particular benefit in this context. Extracellular vesicles (EVs) are a consistent source of tumor-derived RNA due to their prevalence in circulating bodily fluids, the well-established isolation protocols, and the fact that RNA in phospholipid bilayer-enclosed vesicles is protected from blood-borne RNases. Recent results in other types of cancer, including our recent study on plasma EVs from glioblastoma patients suggest that information derived from analysis of EVs from peripheral blood plasma can be integrated in the routine diagnostic tumor approach. In this review, we will examine the diagnostic and prognostic potential of liquid biopsy to detect tumor-derived nucleic acids in circulating EVs from patients with thyroid carcinoma.

## 1. Introduction

Thyroid cancer is the most common malignancy of the endocrine system, representing 3.8% of all new cancer cases in the United States and the ninth most common cancer overall. Its incidence has risen by an average of 5.5% annually over the past ten years [1]. This increase is almost entirely attributed to an increase in papillary thyroid carcinoma (PTC), which comprises 80% of all thyroid cancers [2,3].

Two important diagnostic problems still exist for thyroid cancer that could greatly benefit from the availability of novel blood biomarkers: (i) Distinction between benign and malignant non-functional thyroid nodules. While 3–7% of the world’s population has a palpable nodule, only 5% of those nodules are malignant [2,4]. The initial diagnostic work-up includes the measurement of serum thyroid-stimulating hormone levels, which allows differentiation between functional and non-functional nodules [5]. This is an important characteristic because functional nodules are rarely malignant. If the initial work-up suggests a non-functional nodule with suspicious sonographic features, an ultrasound-guided fine-needle aspirate biopsy should be performed [4]. However, the results of this approach are often inconclusive and may result in over-diagnosis and over-treatment [3]. (ii) Identification of cervical lymph node involvement during staging. Up to 50% of patients with PTC have cervical lymph node involvement. However, preoperative neck ultrasound imaging identifies only half of the lymph nodes that are found during surgery [4].

For the reasons above, liquid biopsy has great potential in the clinical approach to thyroid cancer. It consists of a test done on a sample of blood or, less frequently, other bodily fluids to look for circulating cancer cells, nucleic acids, or proteins from a tumor. Isolation of genetic materials from bodily fluids is a minimally invasive method to diagnose different types of cancer [6]. Thus, the availability of a validated circulating biomarker to distinguish PTCs localized to the thyroid (T+N−) from those with lymph node metastasis (T+N+) would allow to plan ahead for a localized surgery versus a surgery that includes lymphadenectomy.

Extracellular vesicles (EVs) are nano-biological units released from most cell types into the extracellular environment. They include exosomes and ectosomes, distinguished on the basis of their biogenesis [7]. Exosomes are released when multi-vesicular bodies fuse with the plasma membrane, whereas ectosomes bud directly from the plasma membrane. Afterward, they are taken up by neighboring or distant recipient cells. Given the difficulty to isolate pure populations of exosomes or ectosomes, we will refer to them collectively as EVs, as recommended by the recent guidelines set by the International Society for Extracellular Vesicles [8]. EV cargo includes lipids, proteins, and nucleic acids—DNA and several types of RNA. Thus, both mRNA and many types of non-coding RNA, including micro RNA (miRNA), long non-coding RNA (lncRNA), circular RNA (circRNA), small nucleolar RNA (snoRNA), small nuclear RNAs (snRNA), transfer RNA (tRNA), and piwi-interacting RNAs (piRNA), have been reported in EVs [6,9].

Diverse physiological functions of EVs are ascribed to cell-to-cell communication, such as favoring cellular differentiation and epithelial–mesenchymal transition, promoting angiogenesis, and modulating immune responses [10,11]. EVs play also an important role in pathological conditions such as cancer. Thus, the EV-mediated crosstalk between cancerous and non-cancerous cells, e.g., those found in the bone marrow microenvironment, can modulate the biochemistry and consequently the function of stromal components to stimulate the growth, expansion, and spread of cancer cells [12]. Several reports support the concept that EVs also contribute to distant intercellular communication in cancer [13,14]. Recently, Hoshino et al. [15] showed that EV-associated integrins determine organ-specific metastasis through a selective adhesion of EVs to extracellular matrix-enriched cellular areas, followed by their uptake by resident cells at their predicted metastatic destination [11].

Recent results in PTC [16], as well as in other types of cancer [17], suggest that information derived from analysis of EVs from peripheral blood plasma can be integrated in the routine diagnostic approach for patients with non-functional thyroid nodules. Specifically, analysis of the RNA content of EVs in plasma, in addition to being a less invasive approach than tissue biopsy, may have an important role in early detection of cancer and in the distinction of benign from malignant nodules, as well as from localized metastatic thyroid cancer. It can also be utilized to assess the prognosis and detection of progression or response to treatment.

## 2. Liquid Biopsy: History, Advantages

As precision medicine is emerging as the new paradigm in oncology, tumor sampling for molecular characterization is becoming an almost mandatory procedure. However, tissue biopsy is not always feasible and repeated sampling is often impossible due to invasiveness of the procedure. This restricts the use of tissue sampling and molecular characterization only to the diagnostic approach, while therapy monitoring is still impracticable. Liquid biopsy has the potential to overcome this limit, extending the benefits of molecular characterization to early diagnosis and cancer monitoring. This may also produce unprecedented advantages in the treatment of cancer by identifying early events of resistance, relapse, and progression.

Much of the early research on liquid biopsies has been on circulating cell-free DNA (cfDNA) from patients with lung, breast, and prostate cancers. cfDNA was first reported in 1948 when Mandel and Métais demonstrated the presence of both DNA and RNA in plasma from 25 individuals with normal and pathological conditions [18]. Tumor DNA fragments are released into the circulation as a result of programmed cell death or necrosis. cfDNA can be sequenced to help diagnosis, predict relapse, and support clinical decisions for changes in the course of treatment.

The first liquid biopsy test approved by the U.S. Food and Drug Administration for a cancer disease is the blood-based companion diagnostic to select patients with non-small cell lung cancer (NSCLC) for treatment with the epidermal growth factor receptor (EGFR) inhibitor, erlotinib [19]. It can detect gene mutations in EGFR, present in approximately 10–20% of NSCLC patients, who must be identified in order to start an anti-EGFR therapy.

In thyroid cancer, several studies have been conducted on the possible use of cfDNA for prognosis and disease monitoring. For example, the detection of RET M918T in cfDNA has been reported in medullary thyroid carcinoma (MTC) as a specific but not very sensitive event during follow-up [20]. In patients with previously assessed somatic RET mutations, the identification of M918T cfDNA constitutes a prognostic factor for overall survival [18]. While in PTC and MTC the concordance between mutations found in cfDNA and those found within the patient’s surgical specimen does not surpass 50%, higher concordance has been suggested in anaplastic thyroid cancer patients, where tumor growth is much more sustained [21]. Even with this assumption, the present studies on the cfDNA-based liquid biopsy approach suggest that this may not be the best strategy to pursue in the clinics.

Circulating cell-free RNAs, like miRNAs, have also been explored as candidates for liquid biopsy in cancer patients [22]. It has been suggested that miRNAs are actively released in the extracellular fluid and in the bloodstream by viable cancer cells [23]. This is particularly interesting in the case of naïve patients because these biomarkers are abundantly released by viable cells before treatment.

The potential of circulating tumor cells (CTCs) in liquid biopsy has been widely investigated in an increasing number of clinical studies [24], demonstrating the huge interest of both the medical and scientific communities. Early during pathogenesis of a solid tumor, cells are continuously released from the primary tumor and disseminate through the bloodstream. These cells, that may be responsible for metastatic dissemination of the tumor, can be detected very early during tumor development, even before the manifestation of disease symptoms. Different technologies have been developed for the isolation of such a rare cell population, but the most frequently adopted is based on the epithelial immunophenotype of CTCs. While normal mesenchymal cells are quite common in the blood, the presence of cells presenting epithelial epitopes, like EpCAM, may be considered an unusual event, possibly related to cancer cell dissemination [25]. CTCs can be immunomagnetically isolated based on the presence of this epitope with the help of a negative selection such as the common leukocyte antigen CD45. This technique aims to enrich the population of CTCs, thus providing a sort of sampling of the original tumor. The limits of this approach are mainly related to the low number of CTCs, which affect the sensitivity of the approach, and to the inability to detect cells from non-epithelial tumors like soft tissue sarcomas or some non-epithelial ovarian cancers. It has been also demonstrated that the expression of epithelial markers is strongly modulated during the detachment of cells from the tumor, making some epitopes less effective for the isolation of some CTC populations [26]. Publications describing the use of extracellular vesicles, circulating tumor cells, cell-free DNA, and cell-free RNA for diagnosis and monitoring of thyroid cancer are listed in Table 1.

## 3. Potential Advantages of EVs for Liquid Biopsy

The potential of EVs and their content as cancer biomarkers is increasingly being recognized. Tumor-derived EVs may be investigated for their protein expression or genetic profile as diagnostic or prognostic markers [27,28,29,30]. A potential drawback of plasma/serum EV analysis is that they contain not only cancer-derived EVs, but also EVs released by blood cells, endothelial cells, stromal cells, and others. Moreover, during neoplastic growth, immune response and the often associated inflammation may alter the rate of release of EVs. Intriguingly, in a recent study by our group [17], orthotopically growing glioblastoma cells were responsible for 35–50% of all circulating EVs. Although unrelated to thyroid cancer, this study indicates that the fraction of circulating plasma EVs derived from cancer cells is highly significant, even in a disease confined to the brain and relatively isolated from the rest of the body by the blood–brain barrier.

### 3.1. Protein Expression Profile

Protein profiling of EVs is challenging because of their small particle size, low abundance of proteins, and heterogeneity. However, using a laboratory-built high-sensitivity flow cytometer, Tian et al. [31] recently reported a quantitative multiparameter analysis of single EVs down to 40 nm with high analysis rate. By this technique, the authors found a significantly elevated level of CD147-positive EVs in colorectal cancer patients compared to healthy controls, thus indicating potential for future proteomics-based development of cancer diagnostic and therapeutic strategies. In another study, by proteomic analysis of EVs from patients with pancreatic cancer, 18 or 14 proteins were found to be up-regulated and 11 or 14 proteins down-regulated compared with EVs from healthy volunteers or from pancreatitis patients, respectively [32]. Also, studies on colorectal cancer cells and malignant mesothelioma identified specific EV proteins that are considered a potential specific signature for these diseases [33,34].

### 3.2. Non-Coding RNA Content

Since the EV membrane protects RNA from blood-borne RNases and EV-associated RNA is generally free of endogenous RNA contaminants such as ribosomal RNA [35], EVs provide a more consistent source of RNA for disease biomarker detection compared with cellular or free plasma RNA. Interestingly, EV-associated miRNAs remain stable for years when EVs are stored at −20 °C [36]. The presence of functional RNA in EVs was first described in 2006 for murine stem cell-derived EVs [37] and in 2007 for murine mast cell-derived EVs taken up by human mast cells [27]. In ovarian cancer, a specific exosomal signature consisting of 8 miRNAs has been proposed as surrogate diagnostic for cancer screening in asymptomatic subjects [38]. Another study on EVs of patients with melanoma found a correlation between down-regulation of circulating miR-125b and disease progression [39]. The group of miR-1246, miR-3976, miR-4644, and miR-4306 were up-regulated in EVs from 83% of pancreatic adenocarcinoma patients [40]. Dejima et al. discovered that miR-21 and miR-4257 expression in plasma EVs have potential as predictive biomarkers of recurrence in NSCLC patients [41]. Similarly, circulating EV-associated miR-125a-3p in early-stage colon carcinoma [42], as well as miR-320, miR-574-3p, and RNU6-1 in glioblastoma multiforme [43], have been proposed as diagnostic biomarkers for early detection and monitoring of these specific types of cancer. A comprehensive list of EV-associated miRNAs is available in the miRandola database (http://mirandola.iit.cnr.it) [44]. Several studies on esophageal cancer, prostate cancer, and meningioma confirmed the power of EV-associated RNA in cancer diagnosis [45,46,47]. In the FEMX-I melanoma cell line, we recently reported a higher concentration for 49 miRNAs in EVs than in EV-producing cancer cells, including 20 miRNAs with cancer-related function [48]. A correlation of EV-associated integrin α6β4 and integrin α6β1 with lung metastasis, and of integrin αVβ5 with liver metastasis were found by Hoshino et al. [15]. Moreover, novel studies point to the clinical relevance of other types of non-coding RNA as more represented in EVs than miRNAs. Nabet and coll. [49] reported the prevalence of non-coding RNA species distinct from miRNAs, in particular signal recognition particles (SRP) RNA in stromal cell-derived EVs released during co-culture with breast cancer cells. As we discussed in a recent Commentary [50], this is surprising given the general consensus in the EV field on the prevalence of coding RNA and miRNA in EVs. In fact, the main EV data repository, “Vesiclopedia” (https://www.microvesicles.org), contains ~28,000 entries for mRNAs and ~5,000 entries for miRNAs, but no entries for other non-coding RNAs. Undoubtedly, miRNAs have been studied more in depth than other non-coding RNAs in EVs [27,48,51,52,53], but the presence of a multitude of other non-coding RNA families in EVs suggests that other types of EV-associated non-coding RNAs, such as SRP RNA, snRNA, snoRNA, and piRNA [52,54], may have clinical potential as biomarkers for thyroid cancer [55]. More studies have shown the presence of tumor RNA in the plasma/serum of cancer patients [56]. These include mRNAs that are correlated with different tumor genes [57,58,59,60,61], tyrosine kinase mRNA, telomerase components, and viral mRNA. The potential value for therapy management of the EV mRNA profile in patients with metastatic breast cancer has been recently reported [62].

## 4. Potential Advantages of EVs for Liquid Biopsy in Patients with Thyroid Cancer

Although only few studies have been performed on EVs released in the blood by thyroid cancer, recent experimental evidence suggests the involvement of EV miRNAs in thyroid neoplasms, supporting the hypothesis that these non-coding RNAs could be used to develop, refine, or strengthen strategies for diagnosis and management of thyroid cancer. For thyroid cancer patients, EV-based liquid biopsy provides an opportunity to compare a novel tool with the traditional markers used to monitor patients, such as thyroglobulin or calcitonin. While fine-needle aspiration cytology is the gold standard for the differential diagnosis of thyroid nodules [63,64], this procedure has limitations in regard to the discrimination of follicular lesions.

Recent results in other types of cancer, described in the paragraph above, suggest that information derived from analysis of EVs from peripheral blood plasma can be integrated in the routine diagnostic approach to the patient with non-functional thyroid nodules. Moreover, specific alterations of cellular miRNA expression profile have been reported in thyroid carcinoma [65], indicating the possibility that some of these miRNAs, contained in EVs, may be employed as circulating biomarkers. miRNAs in the circulation have been analyzed as potential biomarkers of recurrence in PTC [66]. In many cases in which serum thyroglobulin measurements are difficult to interpret, the analysis of changes in circulating levels of miR-146a-5p and miR-221-3p in PTC patients indicate a good correlation with the American Thyroid Association (ATA)-defined response to therapy classes. Thus, Rosignolo et al. [67] suggested that serum levels of miR-146a-5p and miR-221-3p could be used as complementary biomarkers for the early non-invasive detection of persistent PTC. The association between high circulating levels of miR-146b, miR-222, miR-221, and follicular thyroid proliferation has recently been described [68,69]. Two miRNAs (miR-95, miR-190) were differently expressed in serum of PTC patients. In particular, miR-190 was up-regulated whereas miR-95 was down-regulated, which in combination can be used for the differential diagnosis of thyroid nodules [70].

Other studies have shown that the circulating levels of miR-146b-5p, miR-221-3p, miR-222-3p, and miR-146a-5p were reduced upon tumor excision [67,70,71,72]. The up-regulated expression of miR-146b-5p, miR-221-3p, and miR-222-3p in the circulation of patients with thyroid cancer has also been demonstrated in PTC [73,74], as well as in anaplastic and follicular thyroid carcinoma [75,76] Also, Samsonov et al. [71] found that plasma exosomal miR-21 and miR-181a differentiate follicular from PTC.

An analytical approach employing a miRNA-based assay on thyroid fine needle aspirate smears from routinely prepared cytology slides has recently been proposed to improve the diagnostic process [77]. In addition, we have recently proposed a new miRNA-based molecular classification of PTC [73].

We isolated EVs from blood plasma of patients with PTC before surgery and from age- and sex-matched healthy controls to compare their number and size. Significantly higher numbers of plasma EVs (*p* = 0.025) were found at baseline in thyroid cancer patients (*n* = 6) compared to healthy controls (*n* = 10), as assessed by nanoparticle tracking analysis (Figure 1A). The average size of EVs was similar (Figure 1B). Thus, EV concentration, not their size, helps distinguish PTC patients from healthy controls. We then performed real-time RT-PCR for five putative thyroid cancer-associated miRNAs on EVs isolated from 0.2 mL of plasma of six patients with PTC, and 4 patients with adenoma of the thyroid. Some of the investigated miRNAs showed an increased expression in PTC patients compared to those with adenoma and healthy controls (Figure 1C). This is particularly evident for miR-34a and miR-17-3p, with the first having a significantly different expression in PTC and normal control and the second showing a significantly augmented expression in carcinoma patients if compared to normal controls (*p* = 0.0476). Both miRNAs have been associated with increased proliferation in PTC [101,102]. However, further studies are needed to confirm that these molecules can be valuable liquid biomarkers in thyroid neoplastic diseases.

## Figures and Tables

**Figure 1 ijms-20-01114-f001:**
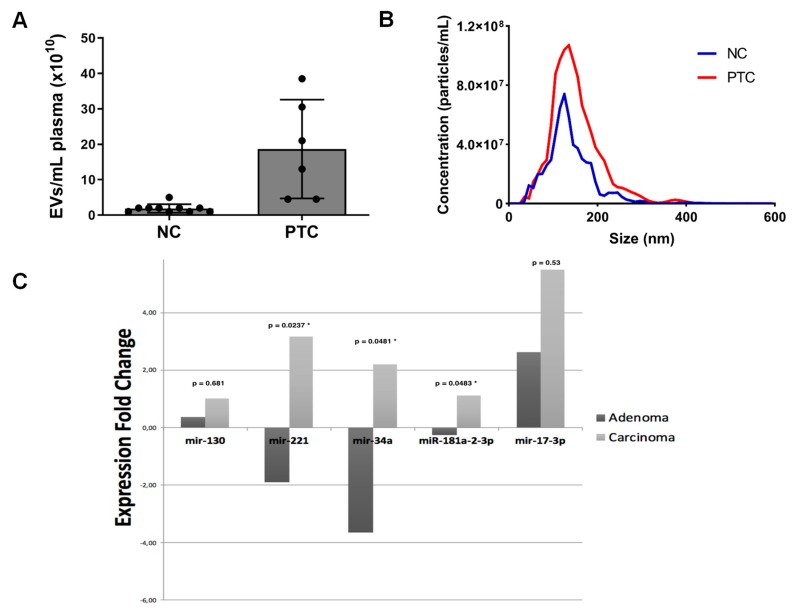
(**A**) Scatter plot of extracellular vesicle (EV) plasma concentration in normal controls (NC) and patients with papillary thyroid carcinoma (PTC) by NTA. EVs were isolated by differential centrifugation. (**B**) Representative size distribution of EVs from plasma of a patient with PTC and a NC. (**C**) Relative quantitation of miRNAs in patients with thyroid adenoma and carcinoma compared to healthy controls, * *p* < 0.01. Patients’ blood samples were collected in disodium EDTA tubes. Plasma samples were prepared by centrifugation and aliquoted into 1.5-mL tubes. EVs were isolated by differential centrifugation as described in Rappa et al. [48]. We measured their plasma concentration and individual size by NTA. Data analysis was performed with NTA 3.0 software. The diffusion coefficient and hydrodynamic radius were determined using the Stokes–Einstein equation, and results displayed as a particle size distribution. Data are presented as the average and standard deviation of six video recordings of 60–90 s per sample. Since NTA is accurate between particle concentrations in the range of 2 × 10^7^ to 2 × 10^9^/mL, samples containing higher numbers of particles were diluted before analysis and the relative concentration calculated according to the dilution factor. Silica microspheres of 100 and 200 nm, supplied by Malvern, were used for calibration.

**Table 1 ijms-20-01114-t001:** Publications describing the use of extracellular vesicles, circulating tumor cells, cell-free DNA, and cell-free RNA for diagnosis and monitoring of thyroid cancer.

Liquid Biopsy in Thyroid Cancer				
Sample Type	Object	Up-/Down-Regulation	Histotype	References
**EV**	miR-146b, miR-222	Up	PTC	[66]
	miR-222, miR-142	Up	PTC	[68]
	miR-25-3p, miR-451a, miR-140-3p, let-7	Up	PTC	[70]
	miR-31-5p, miR-126-3p, miR-145-5p, miR-181a miR-21	Up	PTCFTC	[71]
	miR-21, miR-181a-5p	Up	PTC	[73]
	SRC, TLN1, ITGB2, CAPNS1	-	PTC	[78]
	Drug delivery system	-	TC	[79]
	The increase of EPC-EVs and laminins involves folliculogenesis	Up	FTC	[80]
	lncRNAs, linc-ROR	-	PTC	[74]
	lncRNA MALAT1, SLUG, SOX2,and induced EMT	-	PTC	[81]
**CTC**	Calcitonin-positive CTCs after 12 years	Up	MTC	[82]
	High number of CTCs	Up	DTC DM+	[83]
	CTCs ≥ 5 is worse OS	-	TC	[84]
	High number of CTCs → progressive cancer disease	-	TC	[85]
	PCR detection	-	TC	[86]
**cfDNA/** **ctDNA**	BRAF mutation and deregulation miRNA	Up/Down	PTC	[87]
	RETM91PT mutation	-	MTC	[20]
	cfDNA integrity index	-	TC	[88]
	BRAF mutation	-	TC	[89]
	95% common alteration between cfDNA and tissue DNA	-	FTC	[90]
	BRAF, PIK3CA, NRAS, PTEN, TP53 mutation in cfDNA and tissue DNA	-	ATC	[21]
	BRAF mutation	-	PTC	[91]
	ctDNA panel: 9 cancer gene driver	-	TC	[92]
	BRAFV600	Up	PTC	[93]
	BRAFV600	Up	PTC	[94]
	BRAFV600	Up	DTC	[95]
	cfDNA methylation of β-actin, CDH1,DAPK, CALCA, and RARβ2	-	DTC	[96]
	BRAFV600	Up	PTC	[97]
	High number	Up	DTC	[98]
	BRAFV600	Up	PTC	[99]
**cfRNA**	miR-146a-5p, miR221-3p	Up	PTC	[67]
	let-7e, miR-151-5p, miR-222	Up	PTC	[16]
	miR-579, miR-95, miR-29b, miR-190	Down Up	PTC	[69]
	miR-21, miR-151-5p, miR-222, miR-221	Up	PTC	[72]
	let-7e, miR-151-5p, miR-222	Up	PTC	[16]
	miR-146a-5p, miR-150-5p, miR-199b-3p, miR-342-3p	Down	PTC	[100]

Abbreviations: EV, extracellular vesicles; CTC, circulating tumor cells; cfDNA, cell-free DNA; ctDNA, circulating tumor DNA; cfRNA, cell-free RNA; EPC, endothelial progenitor cells; lncRNA, long non-coding RNA; linc-ROR, long intergenic non-protein coding RNA, regulator of reprogramming; EMT, epithelial-mesenchymal transition; miRNA, micro RNA; PTC, papillary thyroid carcinoma; FTC, follicular thyroid carcinoma; TC, thyroid carcinoma; MTC, medullary thyroid carcinoma; DTC, differentiated thyroid carcinoma; DM+, distant metastasis positive; ATC, anaplastic thyroid carcinoma.

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
