# Peer review of "Extracellular Vesicles from Thyroid Carcinoma: The New Frontier of Liquid Biopsy"

_ijms, 2019, doi:10.3390/ijms20051114_

Reviewer 1 Report

Based on information derived from the analyses of EVs from the plasma of patients with different types of cancers, the authors propose to use circulating EVs for liquid biopsy in patients with thyroid cancer that would help in the identification as well as differentiation between malignant and benign disorder of the thyroid . The authors summarized recent works showing that EVs cargo reflected the state and or progression of neoplastic disease arising from different organs. The data reported by the authors at the end of this review confirm their assumptions and strengthen the validity of their hypotheses.

The review is concise, the results are clear and the reference list is up to date. Minor corrections to this list are recommended (see comments below).

This manuscript would add useful information in the landscape of liquid biopsy and can be accepted for publication in IJMS provided that the authors make the  following  suggested changes.

Comments:

·      In the Introduction, Paragraph 4, in Page 2: ‘’ Given the difficulty to isolate pure populations of exosomes or ectosomes, we will refer to them collectively as EVs.‘’ The authors should  cite the recommendations of the International Society for Extracellular Vesicles recommendations. Authors are invited to cite the following reference: ‘’ Théry C et al. Minimal information for studies of extracellular vesicles 2018 (MISEV2018): a position statement of the International Society for Extracellular Vesicles and update of the MISEV2014 guidelines. J Extracell Vesicles. 2018. 7 (1). doi: 10.1080/20013078.2018.1535750’’.

·      In Page 4: ‘Interestingly, EV-associated miRNAs remain stable for years when EVs are stored at −20°C [29].’’ The reference doesn't seem to mention about EVs storage and miRNA stability. Please verify that the reference is correct.

·      In page 4: ‘’In FEMX-I melanoma, we recently reported a higher concentration for 49 miRNAs in EVs than in EV-producing cancer cells, including 20 miRNAs with cancer-related function [43].’’ Please add FEMX-I melanoma cell line to show that this is an in vitro study.

In Page 4: ‘’ A correlation of EV-associated integrin ?6?4 and integrin ?6?1 with lung metastasis, and of integrin 163 ?V?5 with liver metastasis were found by Hoshino et al. (13).’’ Please put 13 in brackets; .[13].

Author Response

We would like to thank the Reviewer for his/her positive comments. We have made all the recommended changes, as follows:

Point 1:   In the Introduction, Paragraph 4, in Page 2: ‘’ Given the difficulty to isolate pure populations of exosomes or ectosomes, we will refer to them collectively as EVs.‘’ The authors should  cite the recommendations of the International Society for Extracellular Vesicles recommendations. Authors are invited to cite the following reference: ‘’ Théry C et al. Minimal information for studies of extracellular vesicles 2018 (MISEV2018): a position statement of the International Society for Extracellular Vesicles and update of the MISEV2014 guidelines. J Extracell Vesicles. 2018. 7 (1). doi: 10.1080/20013078.2018.1535750’’. 

Response: We have added Thery's MISEV paper.

Point 2:   In Page 4: ‘Interestingly, EV-associated miRNAs remain stable for years when EVs are stored at −20°C [29].’’ The reference doesn't seem to mention about EVs storage and miRNA stability. Please verify that the reference is correct.

Response: We have substituted old Ref. 29 with the paper "Stability of circulating exosomal miRNAs in healthy subjects" by Sanz-Rubio et al. (Sci. Rep. 2018).

Point 3:  In page 4: ‘’In FEMX-I melanoma, we recently reported a higher concentration for 49 miRNAs in EVs than in EV-producing cancer cells, including 20 miRNAs with cancer-related function [43].’’ Please add FEMX-I melanoma cell line to show that this is an in vitro study.

Response: We have added the word "line" after FEMX-I cells on page 4.

Point 4: In Page 4: ‘’ A correlation of EV-associated integrin ?6?4 and integrin ?6?1 with lung metastasis, and of integrin 163 ?V?5 with liver metastasis were found by Hoshino et al. (13).’’ Please put 13 in brackets; .[13].

Response: We have put Ref. 13 in brackets.

Reviewer 2 Report

This review is overall well written and contains the most recent papers about thyroid carcinoma and extracellular vesicles, with a specific focus on liquid biopsy biomarkers. Rappa et al have structured the review in an interesting way, however some minor issues have to be solved before publication.

- An important point is to update the nomenclature and literature about Extracellular vesicles in the manuscript. Authors should include in their manuscript the recent guidelines by the International Society for EVs, MISEV2018 https://www.tandfonline.com/doi/full/10.1080/20013078.2018.1535750 which are a precious and precise document on the most recent discoveries on EVs. Afterwards, authors should introduce the new nomenclature of MISEV guidelines in the introduction of the manuscript and replace the terms exosomes and ectosomes that do not comprehensively describe the heterogeneity and origin of EVs.

- Page 2 line 71 make some examples of cancer models were EVs were studied with reference. The authors could add “Extracellular Vesicles: A New Prospective in Crosstalk between Microenvironment and Stem Cells in Hematological Malignancies.Stem Cells Int. 2018 May 27;2018:9863194. doi: 10.1155/2018/9863194. eCollection 2018. Review.”

- Page 3 line 117 make some examples of cancer models were circulating miRNAs were studied with reference. The authors could add “MicroRNAs as New Biomarkers for Diagnosis and Prognosis, and as Potential Therapeutic Targets in Acute Myeloid Leukemia. Int J Mol Sci. 2018 Feb 3;19(2). pii: E460. doi: 10.3390/ijms19020460. Review”

- The authors should insert in“Potential advantages of EVs for liquid biopsy ” paragraph two subparagraphs: i.e. 3.1 Protein expression profile and add a comment on the “EVs may be investigated for their protein expression profile as diagnostic or prognostic markers.“ (see Characterization and prognostic relevance of circulating microvesicles in chronic lymphocytic leukemia. Leuk Lymphoma. 2017 Jun;58(6):1424-1432. Doi: 10.1080/10428194.2016.1243790; Characterization and proteomic profiling of pancreatic cancer-derived serum exosomes.J Cell Biochem. 2019 Jan;120(1):988-999. doi: 10.1002/jcb.27465; Protein Profiling and Sizing of Extracellular Vesicles from Colorectal Cancer Patients via Flow Cytometry. ACS Nano. 2018 Jan 23;12(1):671-680. doi: 10.1021/acsnano.7b07782; Proteomic Profiling of Exosomes Secreted by Breast Cancer Cells with Varying Metastatic Potential, Proteomics. 2017 Dec;17(23-24). doi10.1002/pmic.201600370. ecc.....) and 3.2 Non coding RNA content.

- Page 5 line 216, in “The up-regulated expression of circulating levels of miR-146b-5p, miR-221-3p, and miR-222-3p in the circulation of patients ...” the words circulating and circulation are repetitive. Modify the text.

- It is well known that there are different methods to isolate EVs from blood and it is more important to specify in the paper the EV purification method. Therefore, in the figure 1A legend, the authors should briefly insert their EV purification method from plasma patients.

- The quality of Figure 1A should be improved.

- In the reference session, the authors should check the references that are underlined ie Ref 18 and 19 and remove the underline.

Author Response

We would like to thank the Reviewer for his/her positive comments and constructive criticisms.

Point 1. An important point is to update the nomenclature and literature about Extracellular vesicles in the manuscript. Authors should include in their manuscript the recent guidelines by the International Society for EVs, MISEV2018 https://www.tandfonline.com/doi/full/10.1080/20013078.2018.1535750 which are a precious and precise document on the most recent discoveries on EVs. Afterwards, authors should introduce the new nomenclature of MISEV guidelines in the introduction of the manuscript and replace the terms exosomes and ectosomes that do not comprehensively describe the heterogeneity and origin of EVs.

Response: We have added MISEV2018 Ref., introduced the new nomenclature in the Introduction and changed Exosomes and ectosomes in EVs where appropriate, as recommended.

Point 2. Page 2 line 71 make some examples of cancer models were EVs were studied with reference. The authors could add “Extracellular Vesicles: A New Prospective in Crosstalk between Microenvironment and Stem Cells in Hematological Malignancies.Stem Cells Int. 2018 May 27;2018:9863194. doi: 10.1155/2018/9863194. eCollection 2018. Review.”

Response: We have added the suggested reference.

Point 3. Page 3 line 117 make some examples of cancer models were circulating miRNAs were studied with reference. The authors could add “MicroRNAs as New Biomarkers for Diagnosis and Prognosis, and as Potential Therapeutic Targets in Acute Myeloid Leukemia. Int J Mol Sci. 2018 Feb 3;19(2). pii: E460. doi: 10.3390/ijms19020460. Review”

Response: We have added the suggested reference.

Point 4. The authors should insert in“Potential advantages of EVs for liquid biopsy ” paragraph two subparagraphs: i.e. 3.1 Protein expression profile and add a comment on the “EVs may be investigated for their protein expression profile as diagnostic or prognostic markers.“ (see Characterization and prognostic relevance of circulating microvesicles in chronic lymphocytic leukemia. Leuk Lymphoma. 2017 Jun;58(6):1424-1432. Doi: 10.1080/10428194.2016.1243790; Characterization and proteomic profiling of pancreatic cancer-derived serum exosomes.J Cell Biochem. 2019 Jan;120(1):988-999. doi: 10.1002/jcb.27465; Protein Profiling and Sizing of Extracellular Vesicles from Colorectal Cancer Patients via Flow Cytometry. ACS Nano. 2018 Jan 23;12(1):671-680. doi: 10.1021/acsnano.7b07782; Proteomic Profiling of Exosomes Secreted by Breast Cancer Cells with Varying Metastatic Potential, Proteomics. 2017 Dec;17(23-24). doi10.1002/pmic.201600370. ecc.....) and 3.2 Non coding RNA content.

Response: We have added the two subparagraphs, the comment and the references, as recommended.

Point 5. Page 5 line 216, in “The up-regulated expression of circulating levels of miR-146b-5p, miR-221-3p, and miR-222-3p in the circulation of patients ...” the words circulating and circulation are repetitive. Modify the text.

Response: The sentence "up-regulated expression of circulating levels of miR-146b-5p, miR-221-3p, and miR-222-3p in the circulation of patients" has been modified as follows: "The up-regulated expression of miR-146b-5p, miR-221-3p, and miR-222-3p in the circulation of patients".

Point 6. It is well known that there are different methods to isolate EVs from blood and it is more important to specify in the paper the EV purification method. Therefore, in the figure 1A legend, the authors should briefly insert their EV purification method from plasma patients.

Response: We have added the EV purification method used, as suggested.

Point 7. The quality of Figure 1A should be improved.

Response: We have improved the quality of Figure 1A.

Point 8. In the reference session, the authors should check the references that are underlined ie Ref 18 and 19 and remove the underline.

Response: We have removed the underline, as recommended.